# Dielectric ordering of water molecules arranged in a dipolar lattice

M. A. Belyanchikov [1✉], M. Savinov[2], Z. V. Bedran[1], P. Bednyakov[2], P. Proschek[3], J. Prokleska[3],
V. A. Abalmasov [4], J. Petzelt[2], E. S. Zhukova [1], V. G. Thomas[5,6], A. Dudka [7], A. Zhugayevych [8],
A. S. Prokhorov[1,9], V. B. Anzin[1,9], R. K. Kremer[10], J. K. H. Fischer [11,12], P. Lunkenheimer [11], A. Loidl [11],
E. Uykur[13], M. Dressel[1,13] & B. Gorshunov [1✉]

Intermolecular hydrogen bonds impede long-range (anti-)ferroelectric order of water. We confine $H_2O$ molecules in nanosized cages formed by ions of a dielectric crystal. Arranging them in channels at a distance of ~5 Å with an interchannel separation of ~10 Å prevents the formation of hydrogen networks while electric dipole-dipole interactions remain effective. Here, we present measurements of the temperature-dependent dielectric permittivity, pyro-current, electric polarization and specific heat that indicate an order-disorder ferroelectric phase transition at $T_0 \approx 3\,K$ in the water dipolar lattice. Ab initio molecular dynamics and classical Monte Carlo simulations reveal that at low temperatures the water molecules form ferroelectric domains in the **ab**-plane that order antiferroelectrically along the channel direction. This way we achieve the long-standing goal of arranging water molecules in polar order. This is not only of high relevance in various natural systems but might open an avenue towards future applications in biocompatible nanoelectronics.

[1] Moscow Institute of Physics and Technology (National Research University), 141701 Dolgoprudny, Moscow Region, Russia. [2] Institute of Physics, Czech Academy of Sciences, 18221 Praha 8, Czech Republic. [3] Department of Condensed Matter Physics, Faculty of Mathematics and Physics, Charles University, 12116 Prague 2, Czech Republic. [4] Institute of Automation and Electrometry SB RAS, 630090 Novosibirsk, Russia. [5] Sobolev Institute of Geology and Mineralogy, RAS, 630090 Novosibirsk, Russia. [6] Novosibirsk State University, 630090 Novosibirsk, Russia. [7] Shubnikov Institute of Crystallography, "Crystallography and Photonics", Russian Academy of Sciences, 119333 Moscow, Russia. [8] Skolkovo Institute of Science and Technology, 143026 Moscow, Russia. [9] Prokhorov General Physics Institute of the Russian Academy of Sciences, Moscow, Russia. [10] Max-Planck-Institut für Festkörperforschung, 70569 Stuttgart, Germany. [11] Experimental Physics V, University of Augsburg, 86135 Augsburg, Germany. [12] T. Kimura Lab, Department of Advanced Materials Science, University of Tokyo, Tokyo, Japan. [13] 1.Physikalisches Institut, Universität Stuttgart, 70569 Stuttgart, Germany. ✉email: belyanchikov@phystech.edu; gorshunov.bp@mipt.ru

For decades intense research activities tackle the question whether water molecules with their rather strong dipole moment of $p_0 = 1.85$ Debye can condense into a ferroelectrically or antiferroelectrically ordered state. In pure liquid water or in $H_2O$ ice no such ordering occurs under ambient conditions or during experimentally available time scales because a complex time-dependent tetrahedral molecular hydrogen-bonded network emerges governed by Pauling's ice rules[1–3]. Proton ordered phases of water ice can be induced by defects introduced via doping with impurities or via irradiation[3,4]. It is suggested that the so-called water ferroelectricity can exist in various natural systems where the role of H-bonds for intermolecular coupling is diminished due to a preferred orientation of water molecules imposed by the host frameworks, thus creating conditions favorable for mutual aligning the $H_2O$ molecular dipoles. In practice, this can be realized when water molecules are adsorbed by two-dimensional interfaces or confined in nanosized channels or cages. Such kind of ferroelectricity of confined water molecules is discussed to be of central importance for a wide range of phenomena in geology, mineralogy, meteorology, soil chemistry, etc. Of special interest is the potential ordering of water molecules in biological systems, where the $H_2O$ molecules are found in fully or partially confined states in cells, membrane channels, and proteins hydration shells[5–8], which can influence the functioning of living organisms substantially.

Various types of dipolar orderings and other exciting properties are predicted based on theoretical analyses or computer simulations of water molecules in nanometer-thick layers on substrates or at the interface[9–13], between two graphene layers[14–17], for $H_2O$ chains within carbon nanotubes[18–23], for networks of $H_2O$ molecules in nanovoids, e.g., in fullerenes[24–26] or within protein hydration shells[27]. Besides fundamental aspects, the interest in such systems is fueled by the perspective that ordered water dipoles can find practical applications in nanoelectronic devices[28–33]. It turned out, however, that it is not so trivial to experimentally verify the predictions made by theory and modeling. Even to realize the dipolar ordering under laboratory conditions appears to be a challenging task. Corresponding reports in the literature are very rare[9–11,34,35], and either observe only a tiny fraction of ordered dipoles ($\leq 1\%$)[10], or raise questions and discussions concerning the reliability of the results[36–41].

An ideal workbench for studying collective effects in ensembles of dipole–dipole coupled water molecules is provided by hydrated dielectric crystals, such as the beryl family. Compared to zeolites and microporous silicates[42,43], the special appeal of these crystals is that they contain just separate $H_2O$ molecules that are isolated within nanosized voids formed by the lattice ions. Being only weakly linked to the ions and separated by 5–10 Å, the water molecules do not experience hydrogen bonding (interaction length 1–2 Å); nevertheless, they interact via long-range dipole–dipole coupling (interaction length 10–100 Å). This kind of network of interacting water molecules is of broad interest and fundamental importance providing the opportunity to study electric-dipolar systems whose properties are expected to be qualitatively different from those occurring in systems with magnetic moments[44,45]. In addition, introducing dopants like Na and K ions at bottlenecks that separate nanovoids can lead to a polarization of $H_2O$ molecules and the formation of statically polarized so-called water-II molecules. Having also the possibility to encapsulate heavy water molecules ($D_2O$, DHO), one gets a unique "laboratory" for studies of various excitations, phases and phase transitions in the water dipolar network. Very recently it was reported[46–51] that $H_2O$ molecules in beryl exhibit the tendency towards a macroscopic alignment of their dipoles: a ferroelectric soft mode was observed with the temperature evolution of dielectric strength and frequency following the Curie–Weiss

and Cochran laws, respectively. However, due to quantum tunneling of the dipoles in the symmetric six-well localizing potential of the hexagonal beryl lattice[52] no phase transition into a macroscopically ordered phase occurs.

In this communication, we present our studies of the dielectric response, pyrocurrent and specific heat of an array of separate $H_2O$ molecules confined within ionic matrix of the orthorhombic cordierite crystal lattice, and supplement our experimental data with Density Functional Theory Molecular Dynamics and Monte Carlo simulations. We discover that at $T_0 \approx 3$ K the water dipolar network undergoes an order–disorder ferroelectric phase transition. In the ground state the water molecules form ferroelectric domains in the **ab**-plane that order antiferroelectrically along the channel direction.

## Results

**Material**. In the present study, we explore the possibility to order water molecules macroscopically by dipole–dipole interaction. To that end, we consider hydrated crystals with a strongly asymmetric localizing potential; i.e. the presence of a certain crystallographic direction along which the molecular dipoles align and form an ordered phase. Crystalline cordierite $(Mg,Fe)_2Al_4Si_5O_{18}$ is ideally suited for these requirements as it is structurally similar to beryl, except for the lack of rotational symmetry. Beryl contains $Si_6O_{18}$ rings, while in cordierite two of six silicon atoms are replaced by aluminum, leading to $(Si,Al)_6O_{18}$ rings that are stacked along the **c**-axis and form channels with 2.5 Å diameter ionic "bottlenecks", see Fig. 1. As a consequence, the cages in cordierite are anisotropic (5.4 Å along the **b**-direction and 6.0 Å along the **a**-axis) and the lattice exhibits orthorhombic symmetry space group $Cccm$[53]. There are indications that the $H_2O$ dipole moments in cordierite are preferably oriented parallel to the **b**-axis[54]. In the present study, slices of well characterized natural hydrous cordierite single crystal were cut along different crystallographic axes in order to measure the dielectric response in all three principal polarizations. We have investigated the electrodynamic properties in a broad frequency range, $\nu = 1$ Hz–3 THz, covering temperatures down to 0.3 K, supplemented by pyroelectric and heat capacity measurements. From the comparison with reference measurements on dehydrated samples, the characteristics of the water molecular subsystem can be extracted.

**Dielectric spectra**. For the E || **a** polarization, we find a broad relaxation-like excitation whose peak frequency decreases when the temperature is lowered down to $T_0 \approx 3$ K and increases on further cooling. Around the same temperature, pronounced anomalies are observed in the dielectric strength of the excitation and of the $H_2O$ contribution to the specific heat. We assign this behavior to an order–disorder phase transition within the water molecular network. Density functional theory molecular dynamics (DFT-MD) and Monte Carlo simulations of water molecules indicate that the low-temperature ordered phase contains domains lying in the **ab**-planes and composed of unidirectional $H_2O$ dipoles polarized preferably along the **b**-axis but ordered antiferroelectrically along the **c**-axis.

Figure 2 presents the temperature-dependent radiofrequency (RF) spectra of the real $\varepsilon'(\nu)$ and imaginary $\varepsilon''(\nu)$ parts of the dielectric permittivity of water molecules in cordierite measured for the polarization E || **a**. Since the MHz–GHz reflectometric technique does not provide absolute values, the corresponding loss data are separately displayed in Supplementary Fig. 1. For the polarization E || **b**, the dielectric response is mostly governed by the higher frequency THz modes and does not reveal features that could be indicative of dipolar ordering. For that reason, in the following we focus on the E || **a** polarization, where strong

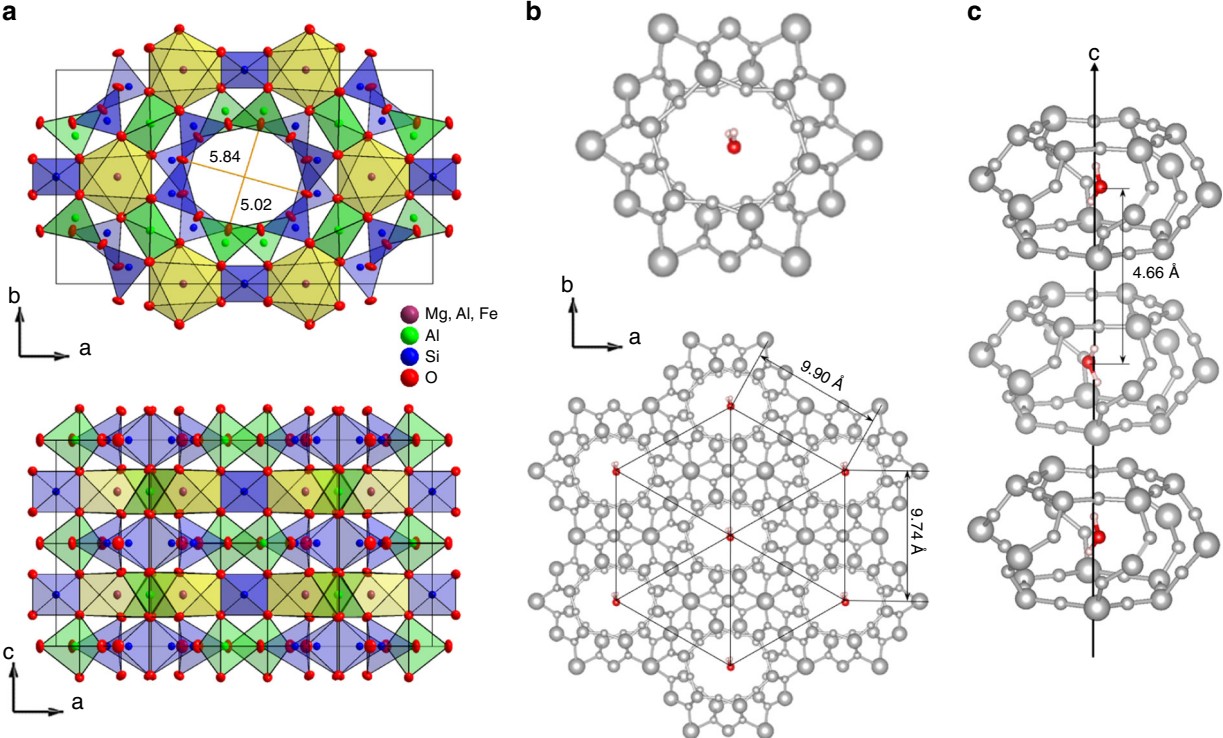

**Fig. 1 Cordierite crystal structure with water molecules within ionic nanopores. a** Unit cell of cordierite crystal from X-ray analysis with thermal ellipsoids at 85 K. **b** The water molecules within cordierite nanopores form two-dimensional triangular lattice within the **ab**-plane. **c** One-dimensional chain of confined water molecules along the **c**-axis.

variations in the spectra are observed on lowering the temperature. We find a pronounced maximum in $\varepsilon'(T)$ with a slope above ≈40 K that follows the Curie–Weiss behavior:

$$\varepsilon'(T) = \varepsilon_{\infty} + C(T - T_c)^{-1}. \quad (1)$$

Here, $\varepsilon_{\infty}$ is the high-frequency dielectric constant, $C$ is the Curie constant and $T_C$ is the Curie temperature. The low-temperature wing of the maximum reveals a pronounced frequency dispersion that is caused by a relaxational band (Fig. 2b, c) signified by a typical relaxation step in $\varepsilon'(\nu)$ together with a broad peak in $\varepsilon''(\nu)$. The fact that this feature is completely absent in water-free crystals unambiguously links the relaxation process to the water subsystem in cordierite. Upon cooling down to $T_0 \approx 3$ K, the band shifts to lower frequencies, mirroring the usual slowing down of thermally activated dipolar dynamics for low temperatures. Surprisingly, a non-canonical increase of peak frequency is observed when cooling further, which implies an acceleration of the involved molecular dynamics. Such behavior is typical for order–disorder ferroelectrics below the phase transition[55].

It should be noted that the detected dielectric relaxation behavior (Fig. 2) in some respects resembles that of so-called relaxor ferroelectrics, which can be regarded as nano-ordered ferroelectrics with a smeared-out diffusive phase transition[56]. There are model simulations predicting typical signatures of relaxors—dispersive maxima due to polar nanoregions in the temperature-dependent dielectric permittivity—in bulk water and in proteins hydration shells[57,58]. Concerning the permittivity, the main distinction between a relaxor and an order–disorder ferroelectric phase transition observed in the present data is the critical behavior of the relaxation time $\tau$ (or critical slowing down of the peak frequency $\nu_p \propto 1/\tau$; Fig. 3), which is typical for order–disorder ferroelectrics. In a relaxor, it

should exhibit glass-like freezing following the Vogel–Fulcher–Tammann law instead[59,60].

## Discussion

The temperature evolution of the detected relaxation process is quantitatively summarized in Fig. 3 by presenting the peak frequency $\nu_p$ and the dielectric strength $\Delta\varepsilon$ of absorption band. At 3 K < T < 8 K, the values of $\Delta\varepsilon$ were determined via fits of the RF spectra of Fig. 2b and c with the Havriliak–Negami function,

$$\varepsilon^*(\nu) = \varepsilon_{\infty} + \Delta\varepsilon\left[1 + (i\omega/\omega_R)^{1-\alpha}\right]^{-\beta}, \quad (2)$$

where $\omega$ is the angular frequency, $\omega_R$ is the angular relaxation frequency and the exponents $\alpha$ and $\beta$ characterize the broadness and asymmetry of the relaxational band, respectively. Fitting results for three selected temperatures are shown in Fig. 2b, c by short-dashed lines. In the lowest temperature range, 0.3 K < $T_0$ < 3 K, the values of $\Delta\varepsilon$ were determined by phenomenologically describing the RF spectra of Fig.2a, b with the sum of two Havriliak–Negami relaxational terms with the dielectric contributions $\Delta\varepsilon_1$ and $\Delta\varepsilon_2$. Around $T_0 \approx 3$ K the peak frequency $\nu_p$ and the relaxation time $\tau = (2\pi\nu_p)^{-1}$ (inset in Fig. 3) exhibit a pronounced minimum and maximum, respectively, which are clear signatures of an order–disorder ferroelectric phase transition[55]. The temperature dependence of the dielectric strength $\Delta\varepsilon$ exhibits a characteristic peak at the transition temperature, again typical for order–disorder ferroelectrics. In addition, the specific heat clearly reveals an anomaly around $T_0$.

Since the electric dipole–dipole interaction energy of water molecules along the channels is $U_{d-d} \approx 22$ meV[47], at room temperature ($k_B T = 26$ meV) the $H_2O$ molecules are mostly decoupled. The situation changes with cooling: the $H_2O$ dimers, trimers, etc., start to form in the channels[47], leading to the Curie–Weiss behavior of Eq. (1) with $C = 390$ K and $T_C = -15$ K (solid blue line in Fig.2a). The negative Curie temperature

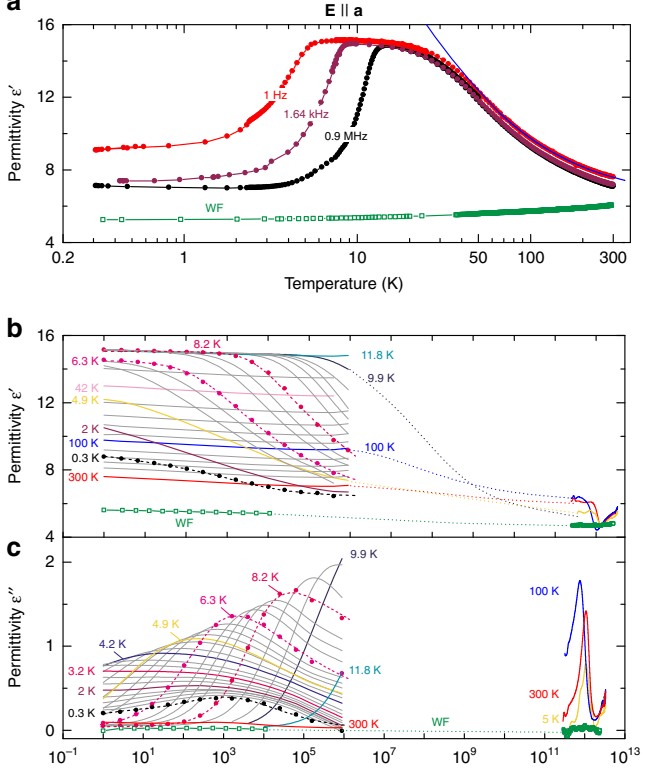

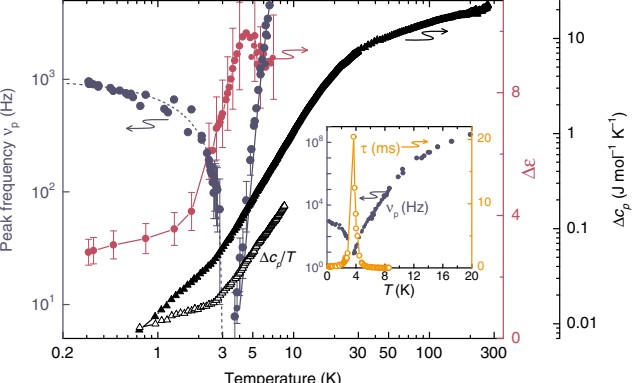

**Fig. 3 Dielectric and thermodynamic indications of a phase transition at $T_0 = 3$ K.** Temperature dependences of peak frequency (dark blue circles) and dielectric strength $\Delta\varepsilon$ (purple circles) of the relaxational excitation (Fig. 2) observed in the hydrous cordierite crystal for the polarization **E** ∥ **a**. The dark blue lines correspond to the fits according to ~ $\exp\{-E_a/k_B T\}^*(T-T_0)$ with $E_a = 6.2$ meV above $T_0$ ($T_0 = 3$ K; solid line) and ~ $(T_0-T)$ below $T_0$ (dashed line), as described in the text. The solid black triangles show the temperature dependence of the excess specific heat of water molecules hosted by cordierite. $\Delta c_p$ is calculated as difference between the specific heat of hydrated and dehydrated crystals. The open black triangles show the specific heat plotted as $\Delta c_p/T$ to stress the low-temperature part. Inset: temperature dependences of the peak frequency $\nu_p$ of the excitation and of the dipole relaxation time $\tau = (2\pi\nu_p)^{-1}$ in a linear temperature scale. The error bars correspond to the ranges of the data that provide satisfactory description of the temperature-dependent relaxational excitation seen in the spectra of the complex permittivity.

**Fig. 2 Temperature evolution of relaxational absorption band.**
**a** Temperature dependences of the real part of the dielectric permittivity $\varepsilon'$ of a hydrous cordierite crystal for **E** ∥ **a** polarization at different frequencies as indicated. In addition, $\varepsilon'(T)$ data for a water-free cordierite crystal (WF, 1.2 kHz) are included. The blue solid line shows the result of a least-square fit with the Curie-Weiss expression, Eq. (1), with $C = 390$ K and $T_C = -15$ K. **b**, **c** Hz–MHz and THz spectra of the real **b** and imaginary **c** parts of the dielectric permittivity of the hydrous cordierite crystal measured at different temperatures as indicated. The short-dashed lines represent least-square fits with Eq. (2) of measured spectra shown by full dots. WF denotes the spectra obtained for a water-free crystal (squares). The dotted lines connecting the radiofrequency and terahertz spectra are guides to the eye.

indicates predominantly antiferroelectric correlations between the dipoles in channels[55,61], as indicated by our DFT-MD and Monte Carlo simulations, see below. Despite strong interactions, we do not observe any detectable signs of a phase transition associated with the appearance of long-range order of the $H_2O$ dipoles at elevated temperatures, most probably due to the incomplete filling (see Sample preparation and characterization section), as vacancies disrupt the one-dimensional water chains.

Below $T \approx 40$ K, however, the thermal energy ($k_B T = 3.5$ meV) becomes comparable to the in-plane dipolar coupling strength $U_{d-d} \approx 3$ meV[47]; deviations from the Curie–Weiss dependence begin to evidence dipolar interactions within the **ab**-planes. At these temperatures the main contribution to the relaxational peak stems from complexes containing water dipoles coupled both along the channels **c**-axis and within the **ab**-planes. Due to polar ordering, the relaxational dynamics of these complexes freezes out at $T_0 \approx 3$ K leading to a strong decrease of the relaxational strength of the band in the ordered state. Concurrently the peak frequency of the band exhibits a V-shaped temperature dependence (Fig. 3). The softening of the peak frequency at 3 K < $T$ < 10 K can be fitted by $\nu_p \sim \exp\{-E_a/k_B T\}^*(T-T_0)$ with $E_a = 6.2$ meV and $T_0 \approx 3$ K (blue solid line in Fig. 3). The Arrhenius factor corresponds to the

non-interacting thermally activated relaxations and the $(T-T_0)$ term describes the critical slowing down of the relaxational response of molecular complexes. At $T < T_0$, the peak frequency follows the $T_0-T$ dependence (blue dashed line in Fig. 3). In all details, the observed features correspond to the behavior of order–disorder ferroelectrics[62–65].

The appearance of an ordered state of $H_2O$ molecular dipoles below $T_0 = 3$ K is conclusively demonstrated by the pyrocurrent that peaks at the same temperature, and the polarization appearing at $T < T_0$, see Fig. 4. The value of low-temperature polarization $P \approx 3$ nC cm$^{-2}$ allows us to estimate the number of dipoles that contribute to this polarization $N = P/p_a \approx 1.4 \times 10^{19}$ cm$^{-3}$. (Here $p_a$ is component of $H_2O$ dipole along the **a**-axis). The obtained value is about two orders of magnitude smaller than the total concentration of water molecules in the crystal $\approx 1.9 \times 10^{21}$ cm$^{-3}$. The reason for this discrepancy lies in the fact that the $H_2O$ dipoles form ferroelectric domains at temperatures below $T_0 = 3$ K but also acquire antiferroelectric order along the **c**-axis. An estimate of the field binding two collinear dipoles separated by a distance $\approx 10$ Å within domain provides a value of 750 kV cm$^{-1}$ [47] that is comparable to crystalline internal fields. Then, the fields of 8 kV cm$^{-1}$ used in our experiments will be able to polarize only small fraction of the dipoles (in the present case about 1%) that are located at the domains boundaries, close to the "defects" (empty cages) or isolated dipoles. We believe that for the same reasons we were not able to detect any meaningful signs of dielectric nonlinearity (hysteresis) in the fields up to 15 kV cm$^{-1}$ because they are also not sufficient to affect the electric coupling between $H_2O$ dipoles.

To provide a more detailed picture of the dipolar ordering, we performed density functional theory molecular dynamics simulations of two $H_2O$ molecules located next to each other along the channel **c**-axis and within the **ab**-plane. The simulations were performed for various temperatures and yield averaged snapshots

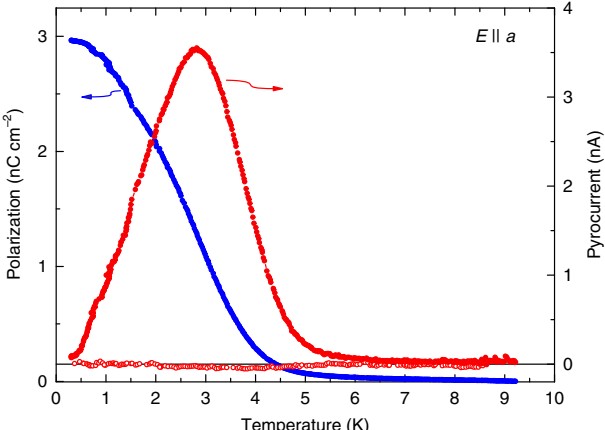

**Fig. 4 Pyrocurrent and polarization indicate a phase transition at $T_0 = 3$ K.** Temperature dependences of pyrocurrent (red) measured for $\mathbf{E} \parallel \mathbf{a}$ polarization while heating in zero electric field after cooling hydrous cordierite crystal in external electric field 8 kV cm$^{-1}$. The temperature-dependent polarization (blue) was calculated from pyrocurrent. Open dots correspond to the pyrocurrent measured during heating in zero electric field after cooling the crystal in zero field.

of the oxygen and two protons of the corresponding two molecules in 1 fs steps during a period of 15 ps (see Supplementary Movie 1–6). Figure 5 displays the **ab**-projection of the positions of oxygen and two protons in two adjacent cages along the channel and in neighboring channels, respectively. At room temperature, hardly any correlations between the sites can be identified, the molecules are nearly independent. Upon cooling, however, the disorder due to thermal motion weakens and the molecules become progressively more confined. The important observation is that the molecules prefer to orient their dipole moments antiparallel within the channels while the dipoles align parallel within the **ab**-planes. Analogous simulations performed for the larger system of four adjacent $H_2O$ molecules in a channel, in plane and occupying all four cages of cordierite crystal unit cell, see Supplementary Figs. 2–4, do not qualitatively change the preferred configurations of dipoles. Only the potentials experienced by the molecules acquire a slightly more complex shape with local minima for oxygen and protons. The results of our Monte Carlo analysis of the dynamics of $N = 3072$ interacting water molecules (see Monte Carlo simulations methods section) are in full agreement with the conclusions drawn from the DFT-MD simulations. As demonstrated in Fig. 6, the additional finding from the Monte Carlo analysis is that in order to minimize the energy of dipolar system, the molecules tend to form in-plane ferroelectric domains with a polarization predominantly along the **b**-axis and alternating its sign along the **c**-direction. We calculated the distribution of domains sizes as a function of their lengths at temperature $T = 0.001$ K (see Supplementary Fig. 5). Domains are considered as arrays of collinear (along the **b**-axis) dipoles with their boundaries given by dipoles of different directions, by defects (empty cages) or by boundaries of the sample. The obtained mean size of the domains is 1.75 and 2.28 lattice distances (see Supplementary Fig. 5) along the **a**-axis and **b**-axis, respectively; after multiplication by the translation lattice vectors the average domains size is 1.496 nm×2.220 nm = 3.32 nm$^2$. Figure 7 displays the temperature dependences of the antiferroelectric order parameter along the **a**-axis and **a**-axis dielectric susceptibility, both clearly demonstrating a phase transition at $T_0 = 3$ K. In other words, our DFT-MD and Monte Carlo results confirm a three-dimensional low-temperature order of the $H_2O$ molecular network. The water molecules in cordierite

form a dipolar lattice with ferroelectric domains in **ab**-planes staggered antiferroelectrically along the channels **c**-axis, as shown in Fig. 8.

Based on our comprehensive experimental and numerical investigations of hydrous cordierite crystals we firmly conclude that $H_2O$ molecules form a highly ordered dipolar lattice at low temperatures. When confining water molecules to the anisotropic nanosized channels composed by the ionic lattice of the crystal, they undergo a ferroelectric order–disorder type phase transition near $T_0 = 3$ K with characteristic features in the temperature dependence of dielectric permittivity, specific heat, pyroelectric current and polarization. Density functional theory molecular dynamics and Monte Carlo simulations indicate the antiferro-electric coupling of $H_2O$ molecular dipoles along the nanochannel **c**-axis and their ferroelectric coupling within the **ab**-planes. As a result, the low-temperature phase is characterized by in-plane ferroelectric domains ordered antiferroelectrically along the nanochannels direction. This long-sought polar phase transition in a system of coupled dipolar water molecules demonstrates that hydrous crystals provide an ideal workbench for studies of different phases and phase transitions in the electric counterpart of magnetic spin systems.

## Methods

**Sample preparation and characterization**. Natural cordierite crystal from India (the detailed location is unknown) have been carefully selected and studied. According to visual inspection with an optical microscope at low magnification (×10), the crystal was free of impurities or foreign phases. 10 independent measurements via microprobe analysis (JEOL JXA-8100, Analytical Center for multielemental and isotope research SB RAS) provided with the following chemical composition in mass %: Na$_2$O—0.204; MgO—13.350; MnO—0.017; K$_2$O—0.008; CaO—0.021; SiO$_2$—49.296; Al$_2$O$_3$—33.140; FeO—1.584; Loss on ignition—2.200 (total—99.821). Recalculation of this chemical analysis on the basis of 18 oxygen atoms per formula gives the following chemical formula:

$(Mg_{1.643}Mn_{0.001}Fe_{0.137}Al_{0.104}Ca_{0.002})_{\Sigma = 1.887}Al_3$ $(Al_{0.921}Si_{5.079})_{\Sigma = 6}O_{18}(H_2O)_{0.756}$ $(Na_{0.041}K_{0.001})_{\Sigma = 0.042}$.

This recalculation assumes that the total weight loss on ignition is due to water loss, which is generally not the case, since some unknown amount of $CO_2$ is also presents in cordierite. Thus, the water content specified in the formula is an estimate from above. From the formula, the filling factors of water-I and water-II[47] can be estimated as ~75.6% and ~4.2%, respectively.

To solve the complex problem of identification of water molecules in the studied cordierite crystal, an accurate X-ray diffraction analysis was carried out. The sets of diffraction reflection intensities at different temperatures were collected on a Rigaku Oxford Diffraction Xcalibur diffractometer equipped with an EOS S2 detector. Our refinement of the cordierite structure provided with the following results. Space group is $Cccm$, $Z = 4$. Maximum angle of scattering of reflections is $\theta = 74.3°$. Unit cell parameters are $a = 17.1004(3)$ Å, $b = 9.7399(2)$ Å, $c = 9.3268$ (2) Å. The residuals that describe the differences between the experimental and modeling data are $R/wR = 1.16/1.35$%. Extremes of the difference Fourier synthesis of electron density are $\Delta\rho_{min}/\Delta\rho_{max} = -0.17/+0.29$ electrons per Å$^3$ for 5693 symmetry-independent reflections. Structural results are presented in Supplementary Table 1. It was found that water molecules are located in the region (0, 0, ¼). The vector connecting protons of the molecule is almost parallel to the crystallographic **c**-axis, and the vector of the molecular dipole moment is slightly tilted relative to the **b**-axis, in agreement with the present DFT-MD results.

**Experimental techniques**. From the crystal, several slices were cut with the **a**, **b**, and **c** crystallographic axes lying within or perpendicular to their planes; these geometries allowed to measure the broad-band dielectric response of the samples in all three principal polarizations of the electric field vector **E** of the probing electromagnetic radiation. For the measurements at frequencies from 1 Hz up to the microwave range, we used a Novocontrol Alpha AN High Performance Frequency Analyzer equipped with a He-flow cryostat JANIS ST-100, an Andeen-Hagerling 2500A capacitance bridge connected to the cryostat utilizing a single-shot 3He insert with embedded coaxial cables, and a coaxial reflectometric technique employing an impedance analyzer Keysight 4991B[66]. A time-domain TeraView 3000 terahertz spectrometer was employed to determine the dielectric spectra up to few terahertz. Dielectric hysteresis loops were studied at frequencies 1–50 Hz using the standard Sawyer-Tower Bridge. The pyrocurrent measurements (0.5–2 K min$^{-1}$ heating after cooling with field up to 8 kV cm$^{-1}$) were realized using the Electrometer High Resistance Meter KEITHLEY 6517B. The polarization was obtained from the pyrocurrent data. The dielectric experiments were complemented by measurements of the heat capacity in the relaxation method employing a PPMS system (Quantum Design). In all experiments, measurements

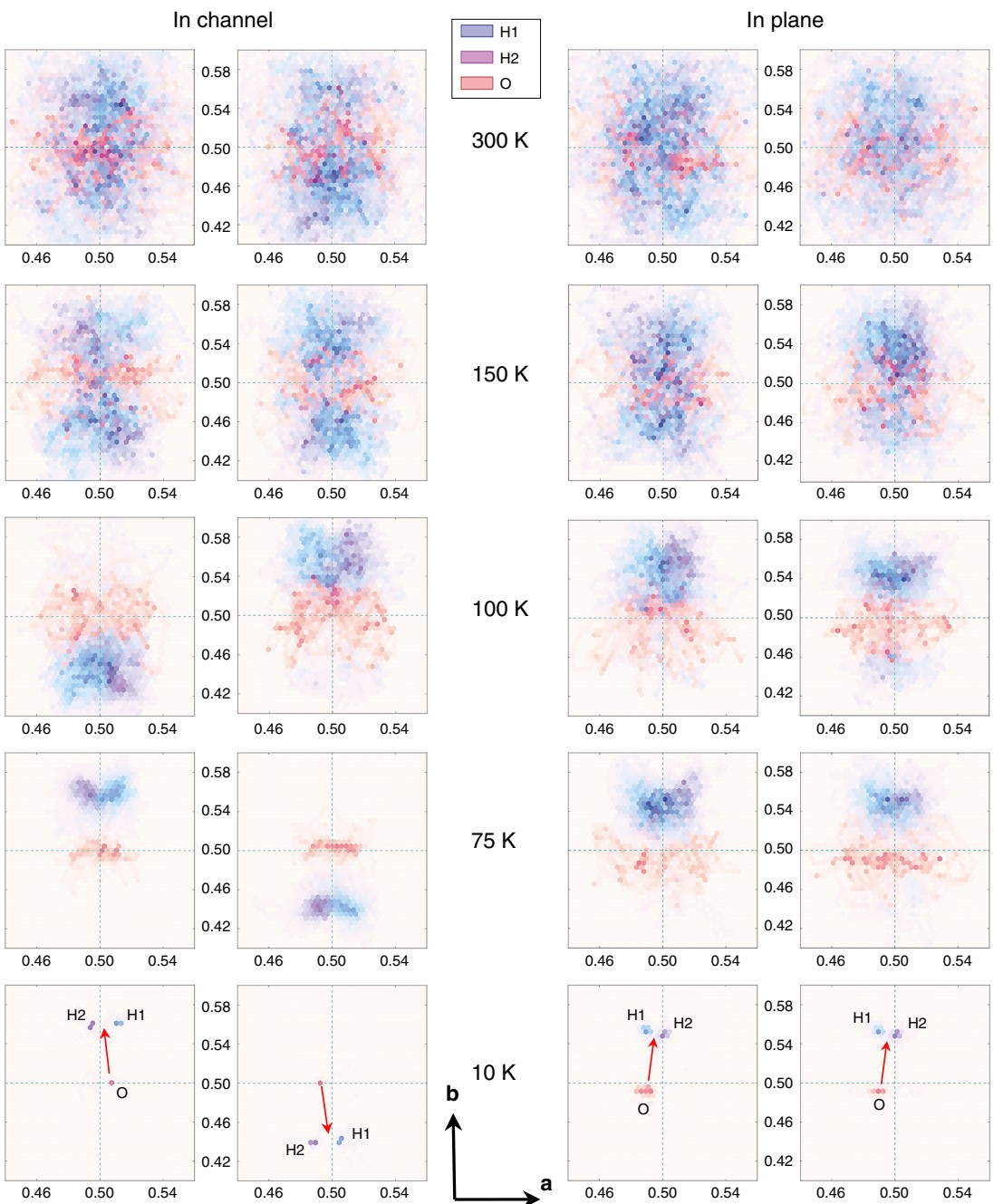

**Fig. 5 Temperature-dependent dynamics of nanoconfined water molecules.** Density functional theory molecular dynamics simulation during 15 ps and at five temperatures of two water molecules within two nanocages of cordierite crystal lattice, located next to each other along the channel **c**-axis (left) and within the **ab**-plane (right). The panels display projections of oxygen ions (O) and two protons (H1, H2) of the $H_2O$ molecules on the **ab**-plane; the **c**-axis is perpendicular to the figure plane. Red arrows represent the dipole moments of the molecules that show antiferroelectric alignment for molecules located along the channel and ferroelectric alignment for molecules in the **ab**-plane. Numbers correspond to the coordinates of atoms within the cages in fractions of translation vectors of cordierite unit cell.

on dehydrated samples (annealing at a temperature 1000 °C for 8 h according to ref. [67]) allowed us to extract the characteristics determined exclusively by water molecules. Complete dehydration was monitored by the weight loss of the specimen and by disappearance of internal vibrations of the $H_2O$ molecule in the infrared spectra.

**Density functional theory molecular dynamics simulation.** Ab initio DFT calculations were carried out using the Vienna Ab initio Simulation Package (VASP)[68,69] with plane wave basis sets cut off at 400 eV, PAW pseudopotentials[70], and Perdew–Burke–Ernzerhof (PBE) exchange-correlation functional[71]. The Brillouin zone integration was carried out using a 1x2x2 **k**-point sampling and Gaussian smearing. One unit cell of cordierite of 116 atoms was simulated with

initial geometry taken from X-ray diffraction (Supplementary Table 1). For all calculations lattice parameters were fixed to the experimentally observed values (see section Sample preparation and characterization above) and periodic boundary conditions were applied. One unit cell of cordierite forms four nanocages, and several types of nanocage filling were simulated. At the first stage, one water molecule and two water molecules within nanocages arranged along the **c**-axis and within **ab**-plane were studied. For each geometry, molecular dynamics simulations were performed at 10 K, 50 K, 75 K, 100 K, 150 K, 300 K for 15 ps or 20 ps with 1 fs time step using Nose–Hoover thermostat. Additionally, molecular analysis was performed for four molecules occupying all four cages of the unit cell and for large system of 1x1x2 and 1x2x1 super cell with four water molecules arranged next to each other along the channel **c**-axis and within the **ab**-plane, respectively. In each procedure, the first 1.5 ps of simulation were considered as thermalization and

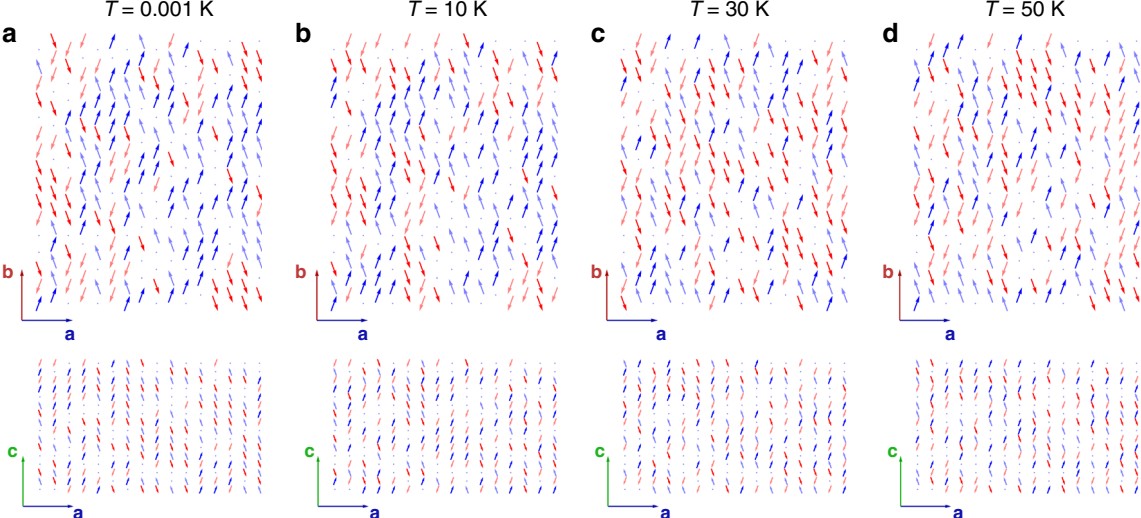

**Fig. 6 Configuration of water molecular dipoles at different temperatures.** We plot the configurations of water molecular dipoles in **ab** and **ac** cross sections at different temperatures obtained from Monte Carlo simulations. Red and blue colors correspond to directions parallel (blue) and antiparallel (red) to the **b**-axis; bright and faded tones correspond to directions along (bright) and against (faded) the **a**-axis. Since the dipoles lie within the **ab**-planes and have zero component along the **c**-axis, arrows in the **ac**-panels are artificially turned to indicate their directions in the **ab**-planes. For the analysis, angles of ±20° relative to the **b**-axis were taken from DFT-MD simulations. Filling factor is 75%.

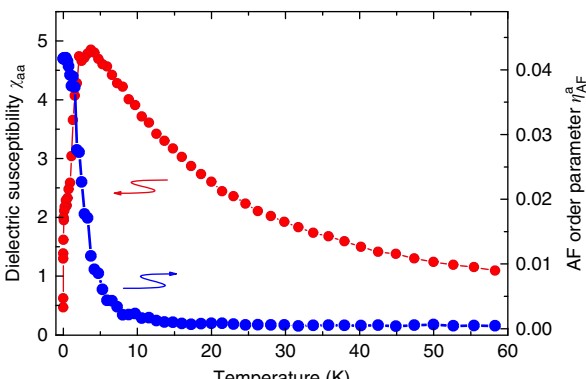

**Fig. 7 Dielectric susceptibility and antiferroelectric order parameter.** Temperature dependences of the dielectric susceptibility (red) and antiferroelectric order parameter (blue) of nanoconfined $H_2O$ molecules along the **a**-axis. The dielectric susceptibility is defined through the polarization $P_a = (1/V) \sum_{i,j,k} p_{ijk}^a$, where $V$ is the crystal volume, the sum is taken over all lattice points $(i, j, k)$ corresponding to the lattice axes (**a, b, c**), $p_{ijk}^a$ is the component of the water dipole $\boldsymbol{p}$ along the **a**-axis at a position $(i, j, k)$, the electric field component $E_a$ along the **a**-axis enters the corresponding expression for dielectric susceptibility via $\chi_{aa} = P_a/\varepsilon_0 E_a$, $\varepsilon_0$ is the permittivity of vacuum. The antiferroelectric order parameter is calculated as $\eta_{AF}^a = (1/N|p^a|) \sum_{i,j,k} (-1)^k p_{ijk}^a$, where $N$ is the number of dipoles in the crystal. The dependences were obtained by Monte Carlo simulations for the electric-dipolar system of water molecules in hydrous cordierite crystal with $N = 3072$ and filling factor 75%.

removed from the subsequent analysis. To find out whether van der Waals (vdW) forces play any significant role in interaction between the water molecules and the ions of nanocage, we performed test calculations using the non-local dispersion corrected vdWDF2 functional[72] (Supplementary Fig. 6, Supplementary Note 1).

**Monte Carlo simulations.** Metropolis algorithm was used to simulate at different temperatures the behavior of the dipole system governed by the dipole–dipole interaction Hamiltonian $H = (8\pi\varepsilon_r\varepsilon_0)^{-1} \sum_{ij} r_{ij}^{-3} \left( \mathbf{p}_i \mathbf{p}_j - 3 \left( \mathbf{p}_i \mathbf{n}_{ij} \right) \left( \mathbf{p}_j \mathbf{n}_{ij} \right) \right)$. Here, $\varepsilon_0$ is the dielectric constant, $\varepsilon_r = 5$ is the relative dielectric permittivity due to other

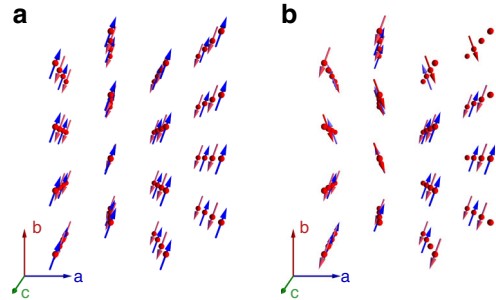

**Fig. 8 Configuration of $H_2O$ dipole moments from Monte Carlo simulations.** We show the configuration of the molecular dipoles of water in cordierite crystalline matrix at zero temperature with two filling factors of water molecules. Red and blue colors correspond to directions along (blue) and against (red) the **b**-axis. **a** 100% filling factor. Dipoles are oriented ferroelectrically in the **ab**-planes and antiferroelectrically along the **c**-axis. **b** 75% filling factor corresponding to the studied hydrous cordierite crystal.

degrees of freedom than the water dipoles, $r_{ij}$ is the distance between two dipoles $\mathbf{p}_i$ and $\mathbf{p}_j$, and $\mathbf{n}_{ij}$ is the unit vector between them. The presented results were obtained for a sample with 16 lattice sites along each axis (a total of 4096 sites) and a 75% dipole filling factor with $N = 3072$ dipoles in total (25% of sites were free of dipoles, we called them defects). Free boundary conditions were applied. These results were well reproduced in simulations with a finite interaction radius cut-off $r_c$ (with up to the 4th next-to-nearest neighbors in the **ab**-plane, which corresponds to 640 lattice sites in the interaction sphere) with periodic boundary conditions. However, in the latter case, the dipoles configuration at the lowest temperature usually consists of domains of radius $r_c$ even in the absence of defects and converges slowly to the ground state. Therefore, we avoided this method in order to see the dipoles configurations in the presence of defects more reliably. The dielectric susceptibility along each axis $\alpha$ was calculated from fluctuations of the average dipole $p_\alpha = N^{-1} \sum_i^N p_{\alpha i}$ as $\chi_\alpha = N p_0^2 (v_0 \varepsilon_0 k_B T)^{-1} \left( \langle p_\alpha^2 \rangle - \langle p_\alpha \rangle^2 \right)$. Here, $N$ is the number of dipoles in a simulation, $p_0 = 1.85$ D is the water molecular dipole, $v_0$ is the volume per dipole, $k_B$ is the Boltzmann constant. For each temperature, the number of Monte Carlo steps per spin was 3500 (the first 500 were attributed to thermalization). The results were averaged over 30 samples with different randomly generated defect configurations. From DFT-MD analysis it was deduced that at low temperatures the dipoles tend to orient at certain angle relative to the **b**-axis. An angle of 20° was taken for all simulations.

## Data availability
The data that support the findings of this study are available from the corresponding author upon reasonable request.

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

## Acknowledgements

The authors acknowledge fruitful discussions with A. Bokov, A. Bush, H. P. Büchler, M. Fyta, V. S. Gorelik, V. Gudkov, J. Hlinka, S. Kamba, G. Shakurov, V. Torgashev. We acknowledge financial support of the Russian Foundation of Basic Research via RFBR grants 18-32-00286 (sample preparation, room temperature experiments) and 18-32-20186 (low-temperature experiments), 20-02-00314 and 18-02-00399 (Monte Carlo simulations), Program 5–100, Czech Science Foundation (Project 20-01527 S), MŠMT Project SOLID21 (CZ.02.1.01/0.0/0.0/16_019/0000760), Deutsche Forschungsgemeinschaft (DFG) via DR228/61-1 and of the Stuttgart/Ulm Research Center for Integrated Quantum Science and Technology (IQST). Low-temperature radio-frequency experiments were performed in MGML (www.mgml.eu), supported within the program of Czech Research Infrastructures (project no. LM2018096). E.U. acknowledges the support of the European Social Fund and by the Ministry of Science Research and the Arts of Baden-Württemberg. A.D. acknowledges support by the Ministry of Science and Higher Education of the Russian Federation (project RFMEFI62119X0035 and the State assignment of the FSRC «Crystallography and Photonics» RAS) and the Shared Research Center FSRC "Crystallography and Photonics", RAS, in part of X-rays diffraction study. We thank G. Siegle and E. Brücher for expert experimental assistance.

## Author contributions

M.A.B., Z.V.B., V.B.A., E.U. carried out the THz experiments; M.S., P.B., P.P., J.P.r. carried out the radiofrequency experiments; J.K.H.F., P.L. carried out the radiofrequency and microwave experiments; M.A.B., Z.V.B., E.S.Z. carried out the infrared experiments; R.K.K. carried out the heat capacity experiments; M.A.B., M.S., M.D., B.G., Z.V.B., V.A.A., P.L., A.L., E.S.Z., A.S.P. analyzed the data; M.A.B., M.S., Z.V.B., P.B., P.P., J.P.r. prepared the samples; V.G.T., A.D. carried out the samples characterization; M.A.B., V.A.A., M.D., B.G., P.L., A.L., J.P.e. carried out the theoretical analysis; M.A.B., V.A.A., A.Z. carried out the numerical simulations; M.A.B., M.D., B.G. conceived and supervised the work; all authors contributed to the manuscript.

## Competing interests

The authors declare no competing interests.
