## [Peer Review File · Nature Communications]

Reviewers' comments:

Reviewer #1 (Remarks to the Author):

Based on experimental and DFT-MD simulations, the authors conclude that the water molecules confined in the Nano channels of cordierite crystal are experienced ferroelectric order-disorder phase transition at 3 K. I have some comments on this work, and the comments are listed below:

1. To let the readers have a clear picture of the structure of the material investigated in this work, the authors should present the molecular structure of the material they employed, especially details of the channels of the structure and how are the water molecules arranged inside the channels?
2. Since the ferroelectric phase transition is observed, can the authors detect the P-E hysteresis loops?
3. In the "Discussion" part, the authors wrote " $T_c = -15$ K". How could the transition temperature below 0 K?
4. The vdW interaction plays a main role between the confined water molecules and the cordierite crystal, thus dispersion corrected density functional should be selected in the DFT computations, e.g. the non-local dispersion corrected vdW-DF2 functional, one of the best functionals to account for the vdW interactions as suggested by previous studies. The authors should at least report a test DFT simulation by using the dispersion corrected functional?
5. As shown in Figure 3, for the "In channel" case, it seems a 180° rotation occurred from 100 K to 75 K. Did that really occur? Are the left picture and the right one reversely placed at 75 K? If it really occurred, the authors should provide a video to show that.
6. Also in Figure 3, I am not fully convinced that the water orientations shown at 10 K depend on the final structure snapshots at 150 K due to the temperatures are too low to make obvious variation of the water orientations at 100 K, 75 K, and 10 K. Are the water orientations always the same at the final time at 150 K? If the water orientations change, would they always align as shown in the figures at 10 K of Figure 3?
7. For the DFT-MD simulations, the authors should include more water molecules. It is too limited to draw the conclusions only based on two water molecules for both the "In channel" and "In plane" case? At least four water molecules are encapsulated at the same time for each case, and the more the better. In addition, please provide a video for each DFT-MD trajectory.
8. What is the reliable evidence for the water dipole orientation as shown in Figure 4b?

Reviewer #2 (Remarks to the Author):

The manuscript titled "Dielectric ordering in dipolar lattice of water in cordierite" reports an order-disorder type phase transition at 3K. The authors attempt to claim that there is ferroelectric H₂O dipole moment by measurements of permittivity and specific heat. The direct evidence of ferroelectric water ice in geometrically confined systems are fundamentally important, which has been extensively studied in theory over past decades as already mentioned by authors. Nevertheless, serious concerns have to be raised in case a wrong conclusion may mislead the readers who are not professional in ferroelectricity, which prevents the publication of this manuscript:

1, the temperature-dependent permittivity demonstrates a giant dielectric loss, which is not observed in the water-free sample. This temperature-dependent anomaly in dielectric constants corresponds to the huge dielectric loss. This is a clear indication that the transition is not ferroelectric transition.

2, the authors claim that there is a relaxation behavior in the frequency-temperature spectra. But this is controversial to the data shown in figure 1 a. We can clearly see that dielectric behavior in 1 kHz measurement are totally different with the ones of 1 Hz, 1.64 kHz, 900 kHz. The 1 kHz data shows a level off of the dielectric constant even at very low temperature (<1 K), which is a hint of possible quantum paraelectricity (refer to the case of SrTiO₃) or extrinsic artifacts. However, this is not shown in 1 Hz, 1.64 kHz, 900 kHz.

3, based on the specific heat measurement, there is a so-called structural transition near 3 K. This is, again, far away from the observation of the dielectric anomaly (5-10 K).

The only two experimental data in this manuscript are not self-consistent. Even so, an anomaly in specific heat or dielectric is just a minor hint of ferroelectricity. In order to make a reasonable argument, pyroelectric measurement and other probe-based techniques should be adopted. It is strongly suggested that the authors should double-check their experimental data and techniques. Although ferroelectricity in water ice is well studied in theory, a direct experimental evidence is important and should be super careful before the publication.

Reviewer #3 (Remarks to the Author):

The authors present an interesting study of the dipolar order of water in cordierite. In particular the authors perform dielectric measurements as a function of frequency and temperature in a hydrated cordierite sample.

Their results indicate the existence of a order-disorder ferroelectric phase transition that occur at T=3 K.

This is quite an experimental achievement, given that the phase transition is due to the ordering of water molecules encapsulated

The study is very interesting and certainly worth publishing in this journal.

However, it needs important modifications in order to make the science and the relevant messages more clear to both the general and field specific audience.

1.- The authors state at the beginning of the introduction that:

"For decades intense research activities tackle the question whether water molecules with their rather strong dipole moment of $p=1.85$ Debye can condense into a ferroelectrically or antiferroelectrically ordered state. In liquid water or in H₂O ice no such ordering occurs because a complex time-dependent tetrahedral molecular hydrogen-bonded network emerges governed by Pauling's ice rules"

However this is not correct. There is a well known order-disorder ferroelectric phase transition in ice-Ih to ice-XI. Details can be found in

Betül Pamuk, Philip B. Allen, and M.-V. Fernández-Serra
Phys. Rev. B 92, 134105 and references there in.

This needs to be changed because right now the manuscript makes incorrect statements.

2.- The paper absolutely needs to have an atomistic picture of water in cordierite. I.e. For full understanding of the ordering it is not enough to provide the sketch in Fig 4b. Indeed this picture is very confusing, and while it can remain, it should be accompanied by an equivalent atomistic picture.

3.- The ab initio results are surprising, but somehow need further explanation.

Here are my questions:

1.- without looking at the atomistic structure of the channel it is difficult to understand why the in plane order should be different from the out of plane order. The authors should provide some insight about why this happens.

Are the 4 nanocages in the unit cell exactly identical?

What is the distance between dipoles in plane versus out of plane?

Why didn't the authors do a simulation with 4 water molecules? This would confirm for sure that the results are consistent with the picture they provide.

2.- With respect to the last point, the authors could show the potential energy landscape of rotating 1 molecule (or dipole) while leaving the other one fixed. This can be done for the 4 molecules in the unit cell, without the need to do ab initio MD simulations. The question this would address is if the coupling between the dipoles is electrostatic or if it needs also to be coupled to some vibrational degree of freedom of the cages.

Only when the authors clarify my questions I will be willing to accept the manuscript for publication. Right now it needs substantial revision.

Reviewer #4 (Remarks to the Author):

The manuscript titled "Dielectric ordering in dipolar lattice of water in cordierite" realized in experiment the long-range dipolar ordering of water, which represents a big step towards achieving the (anti-)ferroelectric control of the confined water phases. This is enabled by confining the water molecules in specially designed nanosized cages, whose dimensions suppress the hydrogen-bonding interaction but still maintain the dipolar interaction between the water molecules. The key evidence of the order-disorder type phase transition comes from the temperature-dependent measurements of the broad-band permittivity and specific heat, showing the transition temperature around 3 K, which is further supported by density functional theory calculations. Those results are very interesting and may trigger further studies on searching for various dipolar ordering of water in confined geometry, which may find practical applications in nanoelectronic devices. I recommend the publication of this work

after the following issues are addressed.

1. It would be better to add a schematic figure in Fig. 1 to show the crystalline cordierite ($(\text{Mg,Fe})_2\text{Al}_4\text{Si}_5\text{O}_{18}$), where the water molecules are confined in the nanosized channels with structural anisotropy.
2. I am not the expert on the dielectric permittivity and specific heat measurement, but it seems that the experiments were delicately performed in a very broad frequency range, $\nu=1 \text{ Hz} - 3 \text{ THz}$, and covering temperatures down to 0.3 K. Here I would like to comment on the DFT-MD simulation part. The authors only simulated two water molecules located next to each other along the channel c-axis, and two water molecules within the ab-plane. However, the real system is long-range ordered and contains much more water molecules. It is well known that three-body systems may behave very differently compared to two-body systems. It would be good to simulate three water molecules in the same way to test the robustness and generality of their conclusion.
3. Page 5, 4th paragraph. It is stated that the ferroelectric orientation is characterized by a slightly lower potential energy (about 3 meV), than the antiferroelectric orientation. Such an energy difference might be too small to stabilize the ferroelectric orientation, even at very low temperature 3 K (the temperature for order-disorder type phase transition). If the energy of ferroelectric and antiferroelectric phases is so close, it would be very difficult to observe the order-disorder type phase transition at 3 K.

Below we present our point-by-point replies to the Referees' comments.

Note that since the submission material should not exceed 150 MB, we were able to provide videos only for a limited number of molecular dynamics along with manuscript. Videos for all other DFT-MD simulations with longer time scale and higher quality are available on Google Drive via the links in Supplementary Information file.

Reviewer #1 (Remarks to the Author):

Based on experimental and DFT-MD simulations, the authors conclude that the water molecules confined in the Nano channels of cordierite crystal are experienced ferroelectric order-disorder phase transition at 3 K. I have some comments on this work, and the comments are listed below:

Referee 1, point 1.

To let the readers have a clear picture of the structure of the material investigated in this work, the authors should present the molecular structure of the material they employed, especially details of the channels of the structure and how are the water molecules arranged inside the channels?

Our reply.

We introduced new Figure 1 that shows in all details the unit cell of cordierite, the water molecules localized inside ionic nanocages and the network of water molecules in cordierite crystalline matrix.

Referee 1, point 2.

Since the ferroelectric phase transition is observed, can the authors detect the P-E hysteresis loops?

Our reply.

For detection of hysteresis loops we used Sawyer-Tower Bridge that operates at frequencies 1 - 50 Hz and is a standard device involved in dielectric studies of "regular" ferroelectrics where the polarization is connected with displacements of crystal lattice ions. We did not observe any sign of dielectric nonlinearity for electric fields up to 15 kV/cm; we believe that these fields available to us are not sufficient to influence the electric coupling between H₂O dipole moments. Indeed, a simple estimate of the field coupling two water dipoles separated by a distance ≈ 5 Å provides [Gorshunov, B. P. *et al.* Incipient ferroelectricity of water molecules confined to nano-channels of beryl. *Nat. Commun.* **7**, 12842 (2016)] a value 6,000 kV/cm, that is typical of crystalline internal fields and exceeds by far the fields needed to shift ions even slightly in regular ionic ferroelectrics. Hence, we are afraid that the observation of the hysteresis loop in the field dependent polarization might be beyond the experimentally accessible range, in general.

Referee 1, point 3.

In the "Discussion" part, the authors wrote " $T_c = -15$ K". How could the transition temperature below 0 K?

Our reply.

The problem that the "transition temperature" obtained by processing the temperature dependence of real permittivity is negative (-15 K) is not relevant since the used Curie-Weiss fit with Eq.1 for frequencies down to 1 Hz is valid only for $T > 40$ K, as seen from Fig. 2a. This means that at these temperatures the dependence of $\epsilon''(T)$ is determined by antiferroelectric correlations between water dipoles in channels, as indicated in the

manuscript. At appropriate place in the revised manuscript we added references to well-known books, [Lines, M. E. & Glass, A. M. Principles and Applications of Ferroelectrics and Related Materials (Clarendon, 1977)] and [R. Blinc & B. Zeks Soft Modes in Ferroelectrics and Antiferroelectrics (North Holland Publishing Co.- Amsterdam) (1974)], where it is described that negative Curie temperature obtained by fitting temperature dependence of real permittivity is an indication of antiferroelectric phase transition. At lower temperatures, below 40 K and closer to $T_0 \approx 3$ K, the type of dynamics of the water dipoles changes due to switching-on of in-plane coupling. Here, at low temperatures, a simple Curie-Weiss law is presumably not valid, and a real phase transition takes place at +3 K.

Referee 1, point 4.

The vdW interaction plays a main role between the confined water molecules and the cordierite crystal, thus dispersion corrected density functional should be selected in the DFT computations, e.g. the non-local dispersion corrected vdWDF2 functional, one of the best functionals to account for the vdW interactions as suggested by previous studies. The authors should at least report a test DFT simulation by using the dispersion corrected functional?

Our reply.

Following the Referee's advice, we have estimated the effect of the van der Waals forces on the interdipole interaction by performing a test DFT simulations using the suggested vdWDF2 functional.

We find that including corresponding corrections does not affect neither the shape of the potential formed by crystal ions nor water molecules dipolar coupling. At the same time, there are slight changes in the potential amplitude that do not lead to any serious qualitative or quantitative changes in our interpretation of phenomena we observe in the water molecular subsystem. Corresponding results are presented in the supplementary information of revised manuscript, section "Dipolar coupling".

Referee 1, point 5.

As shown in Figure 3, for the "In channel" case, it seems a 180° rotation occurred from 100 K to 75 K. Did that really occur? Are the left picture and the right one reversely placed at 75 K? If it really occurred, the authors should provide a video to show that.

Our reply.

The 180 degrees flip of the dipole moment at 100 K did really take place in our simulation. We demonstrate it in Video 3s, shortened version of Supplementary Video 3.

Referee 1, point 6.

Also in Figure 3, I am not fully convinced that the water orientations shown at 10 K depend on the final structure snapshots at 150 K due to the temperatures are too low to make obvious variation of the water orientations at 100 K, 75 K, and 10 K. Are the water orientations always the same at the final time at 150 K? If the water orientations change, would they always align as shown in the figures at 10 K of Figure 3?

Our reply.

Though we are not sure whether we understand correctly the questions raised by the Referee, below we try to address them.

Our DFT-MD simulations done for every selected temperature were independent and performed for each temperature starting with the same initial geometry. The used timescale of 15 ps was long enough for the dynamics at each temperature to become stationary, so that taking five selected temperatures is sufficient to extrapolate the whole temperature dynamics without demanding an *ab initio* MD adiabatic cooling. As is displayed in figure 5 (former figure 3), at 150 K (thermal energy appr. 13 meV) the water molecular dipoles change

orientation with respect to the b axis, but since the estimated antiferroelectric dipolar coupling of about 26 meV twice exceeds the energy of temperature fluctuations, the relative orientation of water molecules is remaining antiferroelectric that can be seen in the figure RL1 below:

Fig.RL1. DFT-MD simulations of variations during 15 ps of the H_2O molecular dipole moments of the two molecules located next to each other in the channel, projected on the a (red color), b (blue color) and c (brown color) axes.

At the same time, hardly any correlations can be seen for the two molecules next to each other in the ab -plane, as seen in Fig.RL2.

Fig.RL2. DFT-MD simulations of variations during 15 ps of the H_2O molecular dipole moments of the two molecules located next to each other in the ab -plane, projected on the a (red color), b (blue color) and c (brown color) axes.

Referee 1, point 7.

For the DFT-MD simulations, the authors should include more water molecules. It is too limited to draw the conclusions only based on two water molecules for both the “In channel” and “In plane” case? At least four water molecules are encapsulated at the same time for each case, and the more the better. In addition, please provide a video for each DFT-MD trajectory.

Our reply.

We agree with the Referee. We have performed DFT-MD simulations with a) four molecules in the unit cell and b) for four molecules in channel and in plane in $1 \times 1 \times 2$ and $1 \times 2 \times 1$ supercell. In addition, we have also done classical Monte Carlo analysis of a system of $N=3072$ interacting dipoles. Both analyses perfectly confirm our original conclusions on

temperature-dependent dynamics and ordering of water molecular dipoles and, furthermore, deliver extra information on the low-temperature dipolar configuration. These results are described in detail in the revised manuscript. Since the submission material should not exceed 150 MB, along with the manuscript we were able to provide only four shortened videos of DFT-MD trajectories (Video 3s, Video 15s, Video 20s, Video 25s). We provide links to videos of all other DFT-MD trajectories in Supplementary Information (Supplementary Video 11-25).

Referee 1, point 8.

What is the reliable evidence for the water dipole orientation as shown in Figure 4b?

Our reply.

Antiferroelectric (oppositely directed) orientation of water molecular dipoles located along the channel *c*-axis, and ferroelectric (collinear) orientation of the dipoles within the *ab*-planes are evidenced by our DFT-MD and our additional Monte-Carlo simulations, as described in the revised manuscript. Perfect match with the measured transition temperature $T_0=3$ K of anomalies in temperature-dependent antiferroelectric (AF) *a*-axis order parameter and dielectric susceptibility obtained by Monte Carlo analysis firmly supports the described configurations of dipoles. Indications of antiferroelectric in-channel dipolar correlations are also provided by negative Curie temperature obtained by Curie-Weiss processing of high-temperature ($T > 40$ K) behavior of real permittivity. Figure 4b of the former version of the manuscript was presented to just schematically show the inter-plane and in-plane types of dipolar coupling. In the revised manuscript, we replace it with the figure that shows more realistic configurations obtained in Monte Carlo simulations. We show that types of in-plane ferroelectric and inter-plane antiferroelectric interactions remain unchanged, but, in addition, there appears domain structure, as described in the revised manuscript.

Referee 2, point 1.

1, the temperature-dependent permittivity demonstrates a giant dielectric loss, which is not observed in the water-free sample. This temperature-dependent anomaly in dielectric constants corresponds to the huge dielectric loss. This is a clear indication that the transition is not ferroelectric transition.

Our reply.

We would not use the expression "giant" to describe the increase of the dielectric loss by a factor of 2. For order-disorder ferroelectrics, for example, the occurrence of a relaxation process with large values of the real part of permittivity, ϵ' , at low frequencies and small ϵ' at high frequencies is a typical property. According to the Kramers-Kronig relations, it is obvious that this sort of dispersion of ϵ' must also lead to a significant dielectric-loss contribution, or, in other words, to a maximum in the dielectric loss ϵ'' spectrum. In various well-established order-disorder ferroelectrics, significant values of the dielectric loss indeed were observed (e.g., [Yamada et al., J. Phys. Soc. Jap. 24, 1053 (1968); Nakamura et al, J. Phys. Soc. Jap. 52, 288 (1983); Takayama et al., J. Phys. Soc. Jap. 53, 4121 (1984)]). Exactly this behavior is observed in our study and demonstrated in Fig.2 (Fig. 1 in the former version of the manuscript). To additionally illustrate Kramers-Kronig consistency between the spectra of real and imaginary permittivities, we add to Fig.2b,c results of least-square fittings of experimental spectra with equation (2), for three selected temperatures. Thus, the detected dielectric loss spectra support the suggested scenario of an order-disorder ferroelectric transition.

Referee 2, point 2.

2, the authors claim that there is a relaxation behavior in the frequency-temperature spectra. But this is controversial to the data shown in figure 1 a. We can clearly see that dielectric behavior in 1 kHz measurement are totally different with the ones of 1 Hz, 1.64 kHz, 900 kHz. The 1 kHz data shows a level off of the dielectric constant even at very low temperature (<1 K), which is a hint of possible quantum paraelectricity (refer to the case of SrTiO₃) or extrinsic artifacts. However, this is not shown in 1 Hz, 1.64 kHz, 900 kHz.

Our reply.

This referee comment seems to be based on a misunderstanding: the 1 kHz curve is for a different material - beryl (as is indicated in Fig. 1a), taken from Gorshunov, B. P. *et al.* Incipient ferroelectricity of water molecules confined to nano-channels of beryl. *Nat. Commun.* **7**, 12842 (2016), that is cited in present manuscript. To avoid confusion and since the dielectric data obtained for beryl is not crucial for presentation of main message of present research, we removed it from the figure.

Referee 2, point 3.

3, based on the specific heat measurement, there is a so-called structural transition near 3 K. This is, again, far away from the observation of the dielectric anomaly (5-10 K).

Our reply.

The anomaly in $\epsilon''(T)$ at "5-10K" that is seen in Fig. 2a, is due to the dipolar relaxation process which is typical for order-disorder ferroelectrics. The temperature position of this relaxation step (in ϵ'') strongly depends on the measurement frequency and clearly does not mark a phase-transition temperature. The anomaly (maximum) in the $\epsilon''(T)$ in Fig. 2a simply corresponds to temperature changes of real permittivity at certain fixed frequency (1 Hz, 1.64 kHz or 0.9 MHz) caused by temperature evolution of relaxation absorption band, when the broad peak in dielectric loss ϵ'' spectrum changes its frequency location while cooling or heating, Fig. 2c. As is well established for order-disorder ferroelectrics, the actual temperature of the ferroelectric transition is revealed in the temperature dependence of parameters of relaxation band, its dielectric contribution $\Delta\epsilon''$ and relaxation time (or peak frequency, that is inverse of relaxation time), as shown in Fig. 3.

Referee 2

The only two experimental data in this manuscript are not self-consistent. Even so, an anomaly in specific heat or dielectric is just a minor hint of ferroelectricity.

Our reply.

Consistency (according to Kramers-Kronig arguments and/or our results of spectral fitting) of the obtained dielectric and specific heat data is explained above in our reply to the first point of Referee 2. As to the low-temperature ordering of water molecular dipoles, it is fully documented by the presented experimental data (including additionally obtained pyrocurrents and polarization) and by DFT and Monte Carlo simulations.

Referee 2.

In order to make a reasonable argument, pyroelectric measurement and other probe-based techniques should be adopted. It is strongly suggested that the authors should double-check their experimental data and techniques. Although ferroelectricity in water ice is well studied in theory, a direct experimental evidence is important and should be super careful before the publication.

Our reply.

We performed additional pyroelectric experiments and clearly detected pronounced anomalies at the phase transition temperature $T_0=3$ K: a sharp peak of pyrocurrent and corresponding electric polarization in field-cooled sample that emerges below 3 K. These experimental results clearly and independently confirm the existence of a phase transition that we detected

with dielectric measurements. The phase transition is also verified by our Monte-Carlo simulations that demonstrate ordering of H₂O dipoles and existence of sharp anomaly in temperature dependences of dielectric permittivity at around 3 K. Corresponding text is added to the revised manuscript.

Reviewer #3 (Remarks to the Author):

The authors present an interesting study of the dipolar order of water in cordierite. In particular the authors perform dielectric measurements as a function of frequency and temperature in a hydrated cordierite sample. Their results indicate the existence of an order-disorder ferroelectric phase transition that occur at T=3 K. This is quite an experimental achievement, given that the phase transition is due to the ordering of water molecules encapsulated. The study is very interesting and certainly worth publishing in this journal. However, it needs important modifications in order to make the science and the relevant messages more clear to both the general and field specific audience.

Referee 3, point 1.

The authors state at the beginning of the introduction that:

"For decades intense research activities tackle the question whether water molecules with their rather strong dipole moment of $p=1.85$ Debye can condense into a ferroelectrically or antiferroelectrically ordered state. In liquid water or in H₂O ice no such ordering occurs because a complex time-dependent tetrahedral molecular hydrogen-bonded network emerges governed by Pauling's ice rules"

However this is not correct. There is a well known order-disorder ferroelectric phase transition in ice-*ih* to ice-*XI*. Details can be found in Betül Pamuk, Philip B. Allen, and M.-V. Fernández-Serra Phys. Rev. B 92, 134105 and references there in. This needs to be changed because right now the manuscript makes incorrect statements.

Our reply.

We agree with the referee and have modified the first paragraph respectively.

Referee 3, point 2.

The paper absolutely needs to have an atomistic picture of water in cordierite. I.e. For full understanding of the ordering it is not enough to provide the sketch in Fig 4b. Indeed this picture is very confusing, and while it can remain, it should be accompanied by an equivalent atomistic picture.

Our reply.

In the revised manuscript we provide an atomistic picture (Fig. 1) of cordierite crystal structure in two main views, with ions, cage size, water molecules and intermolecular distances indicated. In addition, to provide more realistic pictures of dipolar ordering, we show sketch pictures of ordered states with 2D (Fig. 6) and 3D (Fig. 8) snapshots of Monte Carlo simulation.

Referee 3, point 3.

The ab initio results are surprising, but somehow need further explanation.

Here are my questions:

1.- without looking at the atomistic structure of the channel it is difficult to understand why the in plan order should be different from the out of plane order. The authors should provide

some insight about why this happens.

Are the 4 nanocages in the unit cell exactly identical?

What is the distance between dipoles in plane versus out of plane?

Our reply.

As described in our reply to the Referee's previous point, we have provided three additional figures showing atomistic pictures of crystal structure, water molecules and molecular dipoles from which one can see that all nanopores are indeed the same. In the text and in Figure 1, we present distances between neighboring water molecules in *ab*-planes (9.7 – 9.9 Å) and along the *c*-axis (4.66 Å). The distances differ by about a factor of two, and this is the main reason for different interaction strength of the molecules in the *ab*-planes and along the *c*-direction since this distance enters the denominator of the expression describing interdipole coupling. The different character of the in-plane and out-of-plane orders originates from symmetry of water spatial arrangement. In case of out-of-plane *c*-direction, the molecules stack one on top of the other, while molecules located in the *ab*-planes form hexagonal lattice (Figure 1 in the manuscript). The influence of spatial arrangement on the type of dipolar coupling (ferroelectric or anti-ferroelectric) can be estimated from simple electrostatic considerations: rotating dipoles in a 1D system tend to order antiferroelectrically since the sign of coupling between neighbors is negative; in case of hexagonal lattice of rotating dipoles the sign of coupling interaction is positive and minimum energy corresponds to ferroelectric alignment.

Referee 3, point 3.

Why didn't the authors do a simulation with 4 water molecules? This would confirm for sure that the results are consistent with the picture they provide.

Our reply.

The idea behind simulation with two water molecules is to highlight different kinds of interaction in the planes and along the channels. Nonetheless, we agree with the Referee and performed simulations with more water molecules. Please see below our reply to Referee 1, point 7:

We agree with the Referee. We have performed DFT-MD simulations with a) four molecules in the unit cell and b) for four molecules in channel and in plane in 1x1x2 and 1x2x1 supercell. In addition, we have also done classical Monte Carlo analysis of a system of $N=3072$ interacting dipoles. Both analyses perfectly confirm our original conclusions on temperature-dependent dynamics and ordering of water molecular dipoles and, furthermore, deliver extra information on the low-temperature dipolar configuration. These results are described in detail in the revised manuscript. Since the submission material should not exceed 150 MB, along with the manuscript we were able to provide only three shortened videos of DFT-MD trajectories (Video 3s, Video 15s, Video 20s, Video 25s). We provide links to videos of all other DFT-MD trajectories in Supplementary Information (Supplementary Video 11-25).

Referee 3, point 3.

2.- With respect to the last point, the authors could show the potential energy landscape of rotating 1 molecule (or dipole) while leaving the other one fixed. This can be done for the 4 molecules in the unit cell, without the need to do ab initio MD simulations. The question this would address is if the coupling between the dipoles is electrostatic or if it needs also to be coupled to some vibrational degree of freedom of the cages.

Our reply.

Following the Referee's suggestion, we performed simplified study of potential energy landscape for rotational degree of freedom of a water molecule with different filling factors. Corresponding data is included in section "Dipolar coupling" of Supplementary Information. The results is that after subtraction of crystal potential the angular dependence of water

molecules interaction energy perfectly follows the sine function, in excellent agreement with electrostatic expectations.

Only when the authors clarify my questions I will be willing to accept the manuscript for publication. Right now it needs substantial revision.

Reviewer #4 (Remarks to the Author):

The manuscript titled "Dielectric ordering in dipolar lattice of water in cordierite" realized in experiment the long-range dipolar ordering of water, which represents a big step towards achieving the (anti-)ferroelectric control of the confined water phases. This is enabled by confining the water molecules in specially designed nanosized cages, whose dimensions suppress the hydrogen-bonding interaction but still maintain the dipolar interaction between the water molecules. The key evidence of the order-disorder type phase transition comes from the temperature-dependent measurements of the broad-band permittivity and specific heat, showing the transition temperature around 3 K, which is further supported by density functional theory calculations. Those results are very interesting and may trigger further studies on searching for various dipolar ordering of water in confined geometry, which may find practical applications in nanoelectronic devices. I recommend the publication of this work after the following issues are addressed.

Referee 4, point 1.

It would be better to add a schematic figure in Fig. 1 to show the crystalline cordierite $(\text{Mg,Fe})_2\text{Al}_4\text{Si}_5\text{O}_{18}$, where the water molecules are confined in the nanosized channels with structural anisotropy.

Our reply.

In the revised manuscript, we present a new figure 1 where we show detailed structure of cordierite crystal and water molecules that are confined within nanocages.

Referee 4, point 2.

I am not the expert on the dielectric permittivity and specific heat measurement, but it seems that the experiments were delicately performed in a very broad frequency range, $\nu=1$ Hz – 3 THz, and covering temperatures down to 0.3 K. Here I would like to comment on the DFT-MD simulation part. The authors only simulated two water molecules located next to each other along the channel c-axis, and two water molecules within the ab-plane. However, the real system is long-range ordered and contains much more water molecules. It is well known that three-body systems may behave very differently compared to two-body systems. It would be good to simulate three water molecules in the same way to test the robustness and generality of their conclusion.

Our reply.

Below we present our reply to Referee 1, point 7:

We agree with the Referee. We have performed DFT-MD simulations with a) four molecules in the unit cell and b) for four molecules in channel and in plane in $1 \times 1 \times 2$ and $1 \times 2 \times 1$ supercell. In addition, we have also done classical Monte Carlo analysis of a system of $N=3072$ interacting dipoles. Both analyses perfectly confirm our original conclusions on temperature-dependent dynamics and ordering of water molecular dipoles and, furthermore, deliver extra information on the low-temperature dipolar configuration. These results are described in detail in the revised manuscript. Since the submission material should not exceed 150 MB, along with the manuscript we were able to provide only three shortened videos of

DFT-MD trajectories (Video 3s, Video 15s, Video 20s, Video 25s). We provide links to videos of all other DFT-MD trajectories in Supplementary Information (Supplementary Video 11-25).

Referee 4, point3.

Page 5, 4th paragraph. It is stated that the ferroelectric orientation is characterized by a slightly lower potential energy (about 3 meV), than the antiferroelectric orientation. Such an energy difference might be too small to stabilize the ferroelectric orientation, even at very low temperature 3 K (the temperature for order-disorder type phase transition). If the energy of ferroelectric and antiferroelectric phases is so close, it would be very difficult to observe the order-disorder type phase transition at 3 K.

Our reply.

The energy difference of 3 meV corresponds to the temperature of ≈ 35 K, meaning that at temperatures around 3 K the thermal energy will be too low to significantly affect the ferroelectric alignment of water dipoles.

REVIEWER COMMENTS

Reviewer #1 (Remarks to the Author):

The authors addressed our comments mostly well.
Publication is recommended.

Reviewer #2 (Remarks to the Author):

In the revised manuscript, the authors present additional experimental and theoretical data to indicate the observation of dielectric order-disorder transition. The advantage of the nano-cage for the stabilization of dielectric order in H₂O should be highlighted more, comparing to other nanostructures that have been used in previous efforts. This could be the most significant part in this paper. The authors mention that water molecular in channels at a distance of $\sim 5 \text{ \AA}$ with an interchannel separation of $\sim 10 \text{ \AA}$ prevents the formation of hydrogen networks while electric dipole-dipole interactions remain effective. Since there are potentials for bio-compatible nanoelectronic applications, a natural and important question would be what is the size limit for this dielectric order? A size range should be expected in the paper and how to experimentally achieve that? By the size and shape control of the nano-cage or other strategies? By addressing the above concerns, I believe this paper could be considered to be published in Nature communications.

Reviewer #3 (Remarks to the Author):

I find the manuscript almost ready for publication, but I would like the authors to consider my comments below before final publication.

The authors have largely improved the manuscript following the recommendations of the referees. My main concerns were on the simulation side, where I felt that the original manuscript results did not solidly support the conclusions.

The authors have performed new simulations with increased system sizes and checked that indeed the ordering of the molecules is driven by long range electrostatic interactions.

I also liked the addition of fig 2 c, where now the temperature dependent dielectric maximum is fitted to a model which very nicely shows that the phase transition is of order-disorder type. While the authors do not cite this, the behavior is the same of a relaxor ferroelectric. Indeed the authors already say in the new text added right above Fig 4, that there is already some ordering of the water above the transition temperature. I would like the authors to comment on this and connect this to other observations of relaxor ferroelectric behavior in water. Can the authors comment on the possible size of the ordered domains?

Can other type of molecules be in the channels, acting as "dopants" or impurities which lead the different domains?

How is the estimated initial water occupation computed?

I believe that in the section where this is discussed (right above the beginning of the DFT part) this additional discussion is necessary.

With this I feel the manuscript will be ready for publication.

Reviewer #4 (Remarks to the Author):

The authors have nicely addressed my comments as well as those from other reviewers. I recommend the publication in Nature Communications as it is.

We thank the Referees for constructive consideration of all our results and findings. We have considered all Referees' notes and comments and added corresponding changes (highlighted with green color) to the manuscript.

Below we present our point-by-point replies to the Referees' comments.

Reviewer #1 (Remarks to the Author):

The authors addressed our comments mostly well.

Publication is recommended.

Our reply.

We thank the Referee for considerations and comments.

Reviewer #2 (Remarks to the Author):

Referee 2, point 1.

In the revised manuscript, the authors present additional experimental and theoretical data to indicate the observation of dielectric order-disorder transition. The advantage of the nano-cage for the stabilization of dielectric order in H₂O should be highlighted more, comparing to other nanostructures that have been used in previous efforts. This could be the most significant part in this paper.

Our reply.

We have extended the list of objects with confined water molecules and presented references corresponding to their investigations. In the manuscript we now explain why hydrated beryl and cordierite crystals represent model systems for studies of single-particle and collective interactions in an array of dipole-dipole coupled water molecules, making clearer why nano-confinement favors the stabilization of dielectric order. Having discovered incipient ferroelectricity of H₂O molecules in beryl [Gorshunov et al. Nature Communications 7, 12842 (2016)] and an order-disorder phase transition of interacting H₂O molecules in cordierite (present manuscript), we continue research of these systems and are currently preparing for publication our latest results on a quantum phase transition of H₂O molecules in beryl. Studies of more complex systems (H₂O@C₆₀, graphene, zeolites) mentioned in the revised manuscript according to the comment of the Referee are also planned.

Referee 2, point 2.

The authors mention that water molecular in channels at a distance of ~ 5 Å with an interchannel separation of ~ 10 Å prevents the formation of hydrogen networks while electric dipole-dipole interactions remain effective. Since there are potentials for bio-compatible nanoelectronic applications, a natural and important question would be what is the size limit for this dielectric order? A size range should be expected in the paper and how to experimentally achieve that? By the size and shape control of the nano-cage or other strategies? By addressing the above concerns, I believe this paper could be considered to be published in Nature communications.

Our reply.

With the Monte Carlo approach, we calculated the distribution of ferroelectric domains in *ab*-planes as a function of the domains lengths along the *a*-axis and *b*-axis, for the studied sample that has 75% water filling factor. The results are presented in the new Supplementary fig. 5. Based on the obtained distribution, we conclude that at the lowest temperatures of $T=0.001$ K the mean sizes of the domains in the units of the distance between nearest dipoles in the *ab*-plane along each axis (or corresponding lattice distances), are 1.75 and 2.28 along *a*-axis and *b*-axis, respectively. This gives an estimate for the size limit of this dielectric order as requested by the Referee. To experimentally check this, future investigations in other host systems with confinement in this range should be performed. For 100% filling factor, from Monte Carlo

simulations we expect a monodomain state. The corresponding text is added to the manuscript referring to Supplementary Fig. 5 with necessary comments.

Reviewer #3 (Remarks to the Author):

I find the manuscript almost ready for publication, but I would like the authors to consider my comments below before final publication.

The authors have largely improved the manuscript following the recommendations of the referees. My main concerns were on the simulation side, where I felt that the original manuscript results did not solidly support the conclusions. The authors have performed new simulations with increased system sizes and checked that indeed the ordering of the molecules is driven by long range electrostatic interactions. I also liked the addition of fig 2 c, where now the temperature dependent dielectric maximum is fitted to a model which very nicely shows that the phase transition is of order-disorder type.

Referee 3, point 1.

While the authors do not cite this, the behavior is the same of a relaxor ferroelectric. Indeed the authors already say in the new text added right above Fig 4, that there is already some ordering of the water above the transition temperature. I would like the authors to comment on this and connect this to other observations of relaxor ferroelectric behavior in water.

Our reply.

The Referee is right in stating that the detected dielectric relaxation behavior resembles that of a relaxor ferroelectric. Indeed, there are simulations on bulk water or protein hydration shells that predict typical signatures of relaxor behaviors of permittivity - dispersive maxima due to polar nanoregions in temperature-dependent dielectric permittivity; we now cite such papers. It should be noted that the distinction between relaxor ferroelectrics and order-disorder ferroelectrics is rather subtle. The main difference to a relaxor ferroelectric observed in our data is the critical behavior of the peak frequency or the inverse quantity – the relaxation time, which is typical for order-disorder ferroelectrics. In a relaxor, it should exhibit glass-like freezing with the typical Vogel-Fulcher-Tammann law (usually the relaxor community analyzes the peak in $\epsilon''(T)$ seen in relaxors, but it does not make a big difference). A classical critical temperature marking a phase transition does not exist in relaxors. We now mention these facts in the revised manuscript.

Referee 3, point 2.

Can the authors comment on the possible size of the ordered domains?

Our reply.

With the Monte Carlo approach, we calculated the relative number of domains in the ab -planes that contain dipoles collinear along the b -axis dipoles in the ab -planes, as a function of domains lengths along the a - and b -axes. Domains boundaries are represented by dipoles of different directions, by defects or boundaries of the sample. For the lowest temperature of $T=0.001$ K, the mean size of the domains is 1.75 and 2.28 lattice distances along a -axis and b -axis, respectively. Corresponding comments are added to the manuscript referring to Supplementary Fig.5 with necessary explanations.

Referee 3, point 3.

Can other type of molecules be in the channels, acting as "dopants" or impurities which lead the different domains?

Our reply.

In the present manuscript, we study natural cordierite crystals which are known to be a metamorphic rock and usually incorporate H_2O and CO_2 molecules into nanopores, and certain amount of Na and K ions into channels bottlenecks. Alkaline ions can polarize water molecular

dipoles and align them along the *c*-axis; these are so-called water-II molecules. Chemical analysis of the studied crystal provides with an estimate of water-II content of about 4% which we believe is too low to seriously affect the dynamics of the dielectric H₂O dipolar lattice. A certain amount of CO₂ molecules is also expected not to affect the water dipolar lattice since CO₂ molecule does not have persistent dipolar moment.

According to the authors' experience, presently it is not possible to grow in laboratory conditions high-quality single crystals of cordierite with sizes large enough to allow for optical and dielectric experiments. It is thus not possible to study the effect of various impurities and/or inclusions on dynamics of H₂O dipolar system. In this respect, a unique workbench is provided by beryl crystals that can be grown having exceptionally high quality and of centimeters size, in the Novosibirsk laboratory of V.Thomas. Hydrothermal technology allows to introduce into the pores and bottlenecks light and heavy water molecules or alkali ions or engaged, as well as to precisely control their contents. Corresponding opportunities are effectively used by the authors of the present manuscript and corresponding research is in progress.

In principle, considering the Referee's comment in a general aspect, one could think about manipulation with nanoconfinement of other than water polar molecules, species with magnetic moments, or even crystalline frameworks with nanoconfined both, electric and magnetic moments with a possible nanoscale multiferroicity. However, we believe that corresponding task is hardly feasible technically.

We include an appropriate remark in the introduction section.

Referee 3, point 4.

How is the estimated initial water occupation computed?

Our reply.

Since there are no literature data on water molecules distribution within cordierite channels, we believe that it is reasonable to utilize in the Monte Carlo simulations a random filling. This is mentioned in the manuscript in "Monte Carlo simulations" methods section.

I believe that in the section where this is discussed (right above the beginning of the DFT part) this additional discussion is necessary.

Reviewer #4 (Remarks to the Author):

The authors have nicely addressed my comments as well as those from other reviewers. I recommend the publication in Nature Communications as it is.

Our reply.

We thank the Referee for considerations and comments.

REVIEWERS' COMMENTS:

Reviewer #2 (Remarks to the Author):

I suggest the acceptance of this manuscript

Reviewer #3 (Remarks to the Author):

The authors have addressed all my comments and I believe the paper is ready for publication.